A white-box model of S-shaped and double S-shaped single-species population growth

Kalmykov Lev V. 1
Kalmykov Vyacheslav L. 2 3 vyacheslav.l.kalmykov@gmail.com
1 Institute of Theoretical and Experimental Biophysics, Russian Academy of Sciences , Pushchino, Moscow Region , Russian Federation
2 Institute of Cell Biophysics of Russian Academy of Sciences , Pushchino, Moscow Region , Russian Federation
3 Pushchino State Institute of Natural Sciences , Pushchino, Moscow Region , Russian Federation
Dahlem Markus
Electronic publication date: 2015 May 19
Publication date: 2015
Volume: 3
Electronic Location ID: e948
Received 2014 Dec 31; Accepted 2015 Apr 20
Copyright: © 2015 Kalmykov and Kalmykov
Copyright year: 2015
Copyright holder: Kalmykov and Kalmykov
License: This is an open access article distributed under the terms of the Creative Commons Attribution License, which permits unrestricted use, distribution, reproduction and adaptation in any medium and for any purpose provided that it is properly attributed. For attribution, the original author(s), title, publication source (PeerJ) and either DOI or URL of the article must be cited.
License URL: https://creativecommons.org/licenses/by/4.0/

Keywords: Population dynamics, Complex systems, Cellular automata, Individual-based modeling, Population growth curves, Population waves, Artificial intelligence

Funding: The authors declare there was no funding for this work.

==============================
Complex systems may be mechanistically modelled by white-box modeling with using logical deterministic individual-based cellular automata. Mathematical models of complex systems are of three types: black-box (phenomenological), white-box (mechanistic, based on the first principles) and grey-box (mixtures of phenomenological and mechanistic models). Most basic ecological models are of black-box type, including Malthusian, Verhulst, Lotka–Volterra models. In black-box models, the individual-based (mechanistic) mechanisms of population dynamics remain hidden. Here we mechanistically model the S-shaped and double S-shaped population growth of vegetatively propagated rhizomatous lawn grasses. Using purely logical deterministic individual-based cellular automata we create a white-box model. From a general physical standpoint, the vegetative propagation of plants is an analogue of excitation propagation in excitable media. Using the Monte Carlo method, we investigate a role of different initial positioning of an individual in the habitat. We have investigated mechanisms of the single-species population growth limited by habitat size, intraspecific competition, regeneration time and fecundity of individuals in two types of boundary conditions and at two types of fecundity. Besides that, we have compared the S-shaped and J-shaped population growth. We consider this white-box modeling approach as a method of artificial intelligence which works as automatic hyper-logical inference from the first principles of the studied subject. This approach is perspective for direct mechanistic insights into nature of any complex systems.

Introduction

Background

A mechanistic approach corresponds to the classical ideal of science. Existing mathematical approaches to complex systems modeling are rather phenomenological than mechanistic. Ecologists investigate population dynamics phenomenologically, rather than mechanistically (Tilman, 1987). Nonmechanicalness (phenomenologicalness) is still a characteristic for mathematical modeling of complex systems. Most mathematical models in ecology, from simple equations of population growth to complex descriptions of ecosystem dynamics, are not individual-based, i.e., they do not model individuals and their local interactions (Huston, DeAngelis & Post, 1988). The purpose of this study is a mechanistic investigation of the S-shaped and double S-shaped population growth. Individual-based mechanisms of the S-shaped and double S-shaped population growth of vegetatively propagated plants should be completely discrete, logical and consisting of cause–effect and of part-whole relations between micro-subsystems, meso-subsystems and a whole macro-system.

On the types of mathematical models of complex systems

How to create an individual-based mechanistic model of population growth? First, we need to know how to mechanistically model a complex dynamic system. A complex dynamic system may be considered as consisting of interacting subsystems. Interactions between subsystems lead to the emergence of new properties, e.g., of a new pattern formation. Therefore we should define these subsystems and logically describe their interactions in order to create and investigate a mechanistic model. If we want to understand how a complex dynamic system works, we must understand cause–effect relations and part-whole relations in this system. The causes should be sufficient to understand their effects and the parts should be sufficient to understand the whole. There are three types of possible models for complex dynamic systems: black-box, grey-box and white-box models (Fig. 1).

Figure 1 Three types of mathematical models for complex dynamic systems.

This is a schematic representation of a black-box model, a grey-box model and a white-box model with the level of their mechanistic understanding.

Black-box models are completely nonmechanistic. They are phenomenological and ignore a composition and internal structure of a complex system. We cannot investigate interactions of subsystems of such a non-transparent model. A white-box model of complex dynamic system has ‘transparent walls’ and directly shows underlying mechanisms. All events at micro-, meso- and macro-levels of a dynamic system are directly visible at all stages of its white-box model. Unfortunately, in most cases mathematical modelers prefer to use the heavy black-box mathematical methods, which cannot produce mechanistic models of complex dynamic systems in principle. Grey-box models are intermediate and combine black-box and white-box approaches. As a rule, this approach is used in ‘overloaded’ form, what makes it less transparent. Basic ecological models are of black-box type, e.g. Malthusian, Verhulst, Lotka–Volterra models. These models are not individual-based and cannot show features of local interactions of individuals of competing species. That is why they principally cannot provide a mechanistic insight into dynamics of ecosystems. Earlier, we demonstrated that the logical deterministic cellular automata approach allows to create the white-box models of ecosystems with interspecific competition between two, three and four grass species (Kalmykov & Kalmykov, 2013). A similar cellular automata model of interspecific competition was created by Silvertown and colleagues (1992). Their model simulates competitive interactions of five grass species, based on experimentally determined rates of invasion. This is a grey-box model as it is based on stochastic rules of interspecific interactions. Another similar cellular automata model of single plant species was proposed by Komarov and colleagues (2003), where they represented a link between the concept of discrete description of the ontogenesis of plants and the cellular automata. The both two models (Komarov, Palenova & Smirnova, 2003; Silvertown et al., 1992) do not take into account regeneration processes of an ecosystem.

Creation of a white-box model of complex system is associated with the problem of the necessity of an a priori basic knowledge of the modeling subject. The deterministic logical cellular automata are necessary but not sufficient condition of a white-box model. The second necessary prerequisite of a white-box model is the presence of the physical ontology of the object under study. The white-box modeling represents an automatic hyper-logical inference from the first principles because it is completely based on the deterministic logic and axiomatic theory of the subject. The purpose of the white-box modeling is to derive from the basic axioms a more detailed, more concrete mechanistic knowledge about the dynamics of the object under study. We see no other way to obtain a specific and, at the same time, holistic mechanistic understanding of complex systems, apart from the white-box modeling. For providing a strong relevance of our model to the studied problem, we have specified the model’s rules (axioms) strictly in accordance with the subject under study. Each logical rule of the model has a correct ecological and physical interpretation. From an ecological point of view we model a vegetative propagation of rhizomatous lawn grasses. From a physical point of view we model propagation of excitation (autowaves, travelling waves, self-sustaining waves) in an excitable (active) medium. The presence of such physical interpretation makes our specific ecological model more general and more natural. The necessity to formulate an intrinsic axiomatic system of the subject before creating its white-box model distinguishes the cellular automata models of white-box type from cellular automata models based on arbitrary logical rules. If cellular automata rules have not been formulated from the first principles of the subject, then such a model may have a weak relevance to the real problem.

On the white-box modeling of population dynamics

Let’s consider an example of the inadequacy of some ecological models in result of their incompleteness or incorrectness. There are many models of population dynamics that do not take into account what happens with individuals after their death. Dead individuals instantly disappear with roots, stubs, etc. “One reason for the lack of understanding on the part of most botanists results from their failure to take into account the phenomenon of regeneration in plant communities, which was first discussed in general terms by AS Watt in 1947” (Grubb, 1977).

Stephen Hubbell in his Unified Neutral Theory of Biodiversity (UNTB) in fact refuses a mechanistic understanding of interspecific competition: “We no longer need better theories of species coexistence; we need better theories for species presence-absence, relative abundance and persistence times in communities that can be confronted with real data. In short, it is long past time for us to get over our myopic preoccupation with coexistence’ (Hubbell, 2001). However, he admits that ‘the real world is not neutral” (Rosindell et al., 2012). Since the basic postulate of the UNTB about ecological neutrality of the similar species in the ecosystem is wrong, this theory cannot be true. In addition, direct local interactions of individuals are absent in the neutral models in principle. That is why neutral models cannot provide a mechanistic insight into biodiversity. The UNTB models are of black-box and dark grey-box types only— Fig. 1. We agree with James Clark, that the dramatic shift in ecological research to focus on neutrality distracts environmentalists from the study of real biodiversity mechanisms and threats (Clark, 2009). Within the last decade, the neutral theory has become a dominant part of biodiversity science, emerging as one of the concepts most often tested with field data and evaluated with models (Clark, 2009). Neutralists are focused on considering unclear points of the neutral theory—the ecological drift, the link between pattern and process, relations of simplicity and complexity in modeling, the role of stochasticity and others (Rosindell et al., 2012), but not the real biodiversity problems themselves. Attempts to understand neutrality instead of biodiversity understanding look like attempts to explain the obscure by the more obscure. Nonmechanistic ecological models make it difficult to answer basic questions, e.g., Why are there so many closely allied species (Anonymous, 1944)? An example of the difficult ecological discussion is the debates ‘Ecological neutral theory: useful model or statement of ignorance?’ on the forum Cell Press Discussions (Craze, 2012). Understanding of mechanisms of interspecific coexistence is a global research priority. These mechanisms can allow us to efficiently operate in the field of biodiversity conservation. Obviously, such knowledge must be based on mechanistic models of species coexistence. Unfortunately, ecological modelers prefer to use the heaviest black-box mathematical methods, which cannot produce mechanistic models of complex dynamic systems in principle, and not use simple and long-known purely logical deterministic cellular automata, which can produce white-box models and directly obtain clear mechanistic insights into dynamics of complex systems.

Methods

Biological prototype of the model

A vegetative propagation of rhizomatous lawn grasses is the biological prototype of our model (Fig. 3). Festuca rubra trichophylla (Slender creeping red fescue) is the prototype of aggressive vegetative propagation and Poa pratensis L. and Festuca rubra L. ssp. Rubra are the prototypes of moderate vegetative propagation. One individual corresponds to one tiller. A tiller is a minimal semi-autonomous grass shoot that sprouts from the base. Rhizomes are horizontal creeping underground shoots using which plants vegetatively (asexually) propagate. Unlike a root, rhizomes have buds and scaly leaves. One tiller may have maximum three (Fig. 3D) or six rhizoms (Fig. 3B) in the model. Three rhizoms per tiller correspod to moderate propagation (only a half of the nearest microhabitats) and six rhizoms per tiller correspond to aggressive vegetative propagation. A tiller with roots and leaves develops from a bud on the end of the rhizome. A populated microhabitat goes into the regeneration state after an individual’s death. The regeneration state of a site corresponds to the regeneration of microhabitat’s resources including recycling of a dead individual (Fig. 4). All individuals are identical. Propagation of offsprings of one individual leads to colonization of the uniform, homogeneous and limited habitat (Fig. 2 and Movies S5–S8).

Figure 2 S-shaped population growth.

A logical deterministic individual-based cellular automata model of an ecosystem with one species shows both population dynamics and pattern formation. The lattice consists of 25 × 25 sites. Individuals use the hexagonal neighborhood for propagation. The lattice is closed on the torus to avoid boundary effects. (A–C) Population dynamics of the species. S-shaped population growth curve (C). (D–F) Spatio-temporal patterns of the model are represented in numerical form of program implementation.

The cellular automata model

We have used logical deterministic individual-based cellular automata to model the S-shaped population growth mechanistically (Fig. 2). This model demonstrates the underlined individual-based mechanisms. A classical model of the S-shaped population growth is the Verhulst model, which is of completely non-mechanistic black-box nature.

The presented cellular automata model is defined by the 4-tuple:

1. a cellular automata lattice, uniting a collection of sites;

2. a finite set of possible states for each lattice site;

3. a cellular automata neighborhood which consists of a site and its intrinsically defined neighbors;

4. a function of transitions between the states of a lattice site.

The best example of a white-box mechanism is a mechanical watch. Our model metaphorically resembles a mechanical watch in transparent case. A neighborhood logically binds dynamics of all cellular automata sites into one holistic complex dynamic system. There are three most known cellular-automata neighborhoods: von Neumann, Moore and hexagonal. The neighborhood may be of any type. Here we use the hexagonal and tripod neighborhoods which allow to model aggressive and moderate vegetative propagation of rhizomatous lawn grasses (Fig. 3). Different configurations of tripod patterns in Figs. 3C and 3D are results of the fact that the cellular automata neighborhood is implemented successively for each lattice site.

Figure 3 Cellular automata neighborhoods.

A cellular automata neighborhood models a vegetative propagation of plants and defines fecundity and spatial positioning of an individual’s offsprings. Positioning of offsprings is explained by how the cellular automata neighborhood is implemented successively for each lattice site. A central site of the neighborhood is defined by the array element with index (i, j), where i and j are integer numbers. Neighboring sites of the central site are defined by the array elements with indexes. (A) Hexagonal neighborhood. (B) A model example of vegetative propagation of an individual in the hexagonal neighborhood. Offsprings occupy all nearest lattice sites what corresponds to aggressive propagation. A maximum number of offsprings per one individual (fecundity) equals six. (C) Tripod neighborhood. (D) A model example of vegetative propagation of an individual in the tripod neighborhood. Offsprings occupy a half of the nearest lattice sites what corresponds to the moderate propagation. A maximum number of offsprings per one individual equals three.

Integration of reductionist and holistic approaches is one of the challenges for mathematical modeling. Our white-box model of single-species population dynamics opens up new possibilities to solve this challenge. This mechanistic model is hierarchically subdivided into micro-subsystems, meso-subsystems and the whole macro-system. A micro-level is modeled by lattice sites (cellular automata cells). A meso-level of local interactions of micro-objects is modeled by the cellular automata neighborhood. A macro-level is modeled by the entire cellular automata lattice. This is a ‘multy-level’ modelling as parallel logical operations performed on micro-level, meso-level and macro-level of the model. A unique feature of the cellular automata is the possibility to model part-whole relationships mechanistically. The relationships of the parts and the whole are modelled using the transition function (combination of the neigbourhood and rules of transition) between states of a lattice site. Parts are the lattice sites and the whole (ecosystem) is the lattice. On each iteration of evolution of the modeled macrosystem the states of its microsystems are changing simultaneously on the basis of logical ruless taking into account states of the neighbouring microsystems (neigbourhood’s sites). This allows to model how interactions of microsystems (parts) produce evolution of the macro-system (whole) which leads to emergence of its new properties (the ecosystem patterns). The white-box cellular automata model shows interactions of parts within the whole, i.e., ‘part-whole’ relations in the modeled complex system.

Figures 3 and 4 illustrate rules of our model.

Figure 4 Rules of the ecosystem model with one species.

Directed graph of transitions between the states of a lattice site is represented in pictorial (A) and numerical forms (B). The graph represents a birth-death-regeneration process.

Here we show a description of the states of a lattice site (microecosystem) in the single species population growth model. Each site may be in one of the four states 0, 1, 2, 3 (Fig. 4), where:

0—a free microhabitat which can be occupied by an individual of the species;

1—a microhabitat is occupied by a living individual of the species;

2—a regeneration state of a microhabitat after death of an individual of the species;

3—a site in this state represents an element of the boundary that cannot be occupied by an individual.

A free microhabitat is the intrinsic part of environmental resources per one individual and it contains all necessary resources for an individual’s life. A microhabitat is modeled by a lattice site.

The cause–effect relations are logical rules of transitions between the states of a lattice site (Fig. 4B):

0→0, a microhabitat remains free if there is no one living individual in its neighborhood;

0→1, a microhabitat will be occupied by an individual of the species if there is at least one individual in its neighborhood;

1→2, after death of an individual of the species its microhabitat goes into the regeneration state;

2→0, after the regeneration state a microhabitat will be free if there is no one living individual in its neighborhood;

2→1, after the regeneration state a microhabitat will be occupied by an individual of the species if there is at least one individual in its neighborhood;

3→3, a site remains in this state, which defines a boundary site.

These logical statements are realized for all micro-levels (sites) with their meso-levels (neighborhoods) and thus for the whole macro-level (lattice) of the complex system on each time iteration. We consider implementation of this algorithm as hyper-logical operations or automatic hyper-logical inference from the first principles of the studied subject.

Results and Discussion

According to Watt (1947), a plant ecosystem may be considered ‘as a working mechanism’ which ‘maintains and regenerates itself.’ Our model demonstrates a such mechanism. From a more general physical point of view we model an active (excitable) media with autowaves (travelling waves, self-sustaining waves) (Kalmykov & Kalmykov, 2013; Krinsky, 1984; Zaikin & Zhabotinsky, 1970). Active medium is a medium that contains distributed resources for maintenance of autowave propagation. An autowave is a self-organizing and self-sustaining dissipative structure. An active medium may be capable to regenerate its properties after local dissipation of resources. In our model, propagation of individuals occurs in the form of population waves. We use the axiomatic formalism of Wiener & Rosenblueth (1946) for modeling of excitation propagation in active media. In accordance with this formalism rest, excitation and refractoriness are the three successive states. In our formalism the rest state corresponds to the free state of a microhabitat, the excitation state corresponds to the life activity of an individual in a microhabitat and the refractory state corresponds to the regeneration state of a microhabitat. All states have identical duration. If the refractory period will be much longer than the active period, then such a model may be interpreted, for example, as propagation of the single wave of dry grass fire. Time duration of the basic states can be easily varied using additional states of the lattice sites.

Different initial conditions may lead to formation of different spatio-temporal patterns and as a result they may lead to different dynamics of the system. Using the Monte Carlo method, we have investigated the influence of different initial conditions on population dynamics of one species. We have investigated two different boundary conditions, two different cellular automata neighborhoods and four different lattice sizes (Figs. 5 and 6). Ecosystem dynamics on the plane with boundary is more natural than on a torus, where boundary effects are absent. The models with non-periodic boundary conditions correspond to laboratory and field experiments where experimental plots also have a boundary. Models with periodic boundary conditions are investigated more commonly, as they allow to avoid boundary effects. Periodic boundary conditions cannot be reproduced in real ecosystems, but they allow to investigate models in a more general form. Therefore, we decided to explore the both types of boundary conditions. Figure 5 shows the results obtained in the study of aggressive propagation and Fig. 6 shows the results obtained in the study of moderate propagation. In Figs. 5B–5D and 6E–6H we show the S-shaped population growth and in Figs. 6B–6D we show the double S-shaped population growth. Sizes of the lattice which define available space for colonization consisted of 3 × 3, 8 × 8, 23 × 23 and 98 × 98 sites. We have investigated the boundary conditions of two types—when the lattice was closed on the torus by periodic conditions (Figs. 5A–5D and 6A–6D) and when the lattice has a boundary (Figs. 5E–5H and 6E–6H). There were no changes of population dynamics in result of the different initial positioning of an individual on the lattice in cases with periodic boundary conditions (Figs. 5A–5D and Figs. 6A–6D). In cases when the lattice has a boundary, different initial positioning of an individual lead to differences in population dynamics (Figs. 5E–5H and 6E–6H). Moreover, increasing of the lattice may lead to more complex dynamics (Figs. 5E–5H and 6E–6H).

Figure 5 Results of the Monte Carlo simulations with the hexagonal neighborhood.

Investigation of the influence of boundary conditions, initial positioning of an individual and lattice sizes on single-species population dynamics. (A–D) The lattice is closed on the torus to avoid boundary effects. (E–H) The lattice has a boundary. Every Monte Carlo simulation consisted of 100 repeated experiments with different initial positioning of an individual on the lattice.

Figure 6 Results of the Monte Carlo simulations with the tripod neighborhood.

Investigation of the influence of boundary conditions, initial positioning of an individual and lattice sizes on single-species population dynamics. (A–D) The lattice is closed on the torus to avoid boundary effects. (E–H) The lattice has a boundary. Every Monte Carlo simulation consisted of 100 repeated experiments with different initial positioning of an individual on the lattice.

Periodic fluctuations in numbers of individuals are observed at the plateau phase in most of the experiments. With increasing of the lattice size, these periodic fluctuations in population size become less visible. The periodic fluctuations on the plateau phase are absent when the lattice consists of 3 × 3 sites in the case of the tripod neighborhood (Figs. 6A and 6E). The similar plateau phases without fluctuations were found at the 3N × 3N sizes of the lattice (6 × 6, 9 × 9, 12 × 12, 15 × 15, 18 × 18, 27 × 27 lattices were tested), with and without boundary effects and when the neighborhood was tripod.

We show four Movies S1–S4 as examples of the Monte Carlo simulations. Each Monte Carlo simulation consists of five repeated experiments with different initial positioning of an individual on the lattice. The lattices are homogeneous and limited in all experiments. They consist of 23 × 23 sites available for occupation by individuals. In Movie S1 the lattice is closed on the torus and the neighborhood is hexagonal. In Movie S2 the lattice has a boundary and the neighborhood is hexagonal. In Movie S3 the lattice is closed on the torus and the neighborhood is tripod. In Movie S4 the lattice has a boundary and the neighborhood is tripod.

In more detail individual-based mechanisms of the double S-shaped population growth curve are presented in Fig. 7C and Movie S7. Details of individual-based mechanisms of three types of the S-shaped population growth curves are presented in Figs. 7A, 7B and 7D and in Movies S5, S6 and S8.

Figure 7 Population growth curves.

The lattice size which is available for occupation consisted of 50 × 50 sites in all four cases. (A) S-shaped curve with short phase of decelerating growth. Cellular automata neighborhood is hexagonal and the lattice is closed on the torus (Movie S5). (B) S-shaped curve with sharp transition to long phase of decelerating growth. Cellular automata neighborhood is hexagonal and the lattice has a boundary (Movie S6). (C) Double S-shaped population growth curve. Cellular automata neighborhood is tripod and the lattice is closed on the torus (Movie S7). (D) S-shaped curve with very long phase of decelerating growth. Cellular automata neighborhood is tripod and the lattice has a boundary (Movie S8).

Figure 7A shows the S-shaped population growth curve with short phase of decelerating growth. This curve reaches a plateau earlier than on population curves in Figs. 7B–7D. The plateau is reached on the 32nd iteration (Movie S5). The higher rate of population growth is explained by aggressive propagation and by the lack of boundary effects because the lattice is closed into a torus.

Figure 7B shows the S-shaped population growth curve with sharp transition to long phase of decelerating growth. This curve has a sharp slowdown of population growth before the beginning of phase of decelerating growth. It occurs on the 25th iteration, when population waves of aggressively propagating species reach the habitat boundary (Movie S6). In contrast to the curve in Fig. 7A, this population curve reaches the plateau on the 49th iteration. Reduced population growth rate of aggressively propagating species is explained by the presence of boundary effects because the lattice has a boundary.

In Fig. 7C the population growth curve has a double S-shaped form. The double S-shaped population growth is a result of temporary slowdown of growth, which occurs at the stage when colonization of the free field is replaced by interpenetration of colliding population waves into already occupied areas. Starting from the 34th iteration, the stage of gradually compaction of populated areas begins (Movie S7). This compaction arises from the fact that after rounding of the torus population waves occupy the remaining free sites in the partially populated part of habitat as result of a ‘phase shift’ of the colliding waves. The free vacancies in population waves remain in result of moderate propagation of individuals. The moderate propagation is modeled by the tripod neighborhood. Speed of the sealing colonization increases slowly due to the form of the population waves which invade into already occupied areas by the expanding wedge. At the same time, a contribution into population growth from colonization of the areas which consist only of free sites (microhabitats) decreases. The areas which consist only of free microhabitats disappear on the 49th iteration. The population growth rate temporarily slows down what forms the first plateau of the curve. This plateau phase lasts during 5 iterations. The accelerating of additional compactization of population waves leads to the new population growth starting from the 53rd iteration. The population curve reaches the second plateau on the 98th iteration.

Figure 7D shows the S-shaped population growth curve with very long phase of decelerating growth. This curve reaches a plateau on the 72nd iteration (Movie S8). Reduced population growth rate and reduced maximum number of individuals in the habitat (834 individuals) are a result of the boundary conditions and the moderate fecundity of individuals (because of the tripod neighborhood).

The S-shaped and the J-shaped population growth curves

We have investigated the S-shaped population growth which is limited by following factors: finite size of the habitat (limited resources), habitats’ size, type of boundary conditions of habitat, intraspecific competition, lifetime of individuals, regeneration time of microhabitats, fecundity of individuals (Figs. 2, 5–7 and Movies S1–S8). In this section, we show the model of the J-shaped population growth and investigate two cases of geometric population growth. Unlike of the S-shaped population model, the J-shaped population model describes a situation in which population growth is not limited in resources, by intraspecific competition or for any other environmental reasons. The J-shaped population model describes a full reproductive potential which lead to geometrical population growth (Fig. 8). In other respects this model is similar to our model of the S-shaped population growth. It also takes into account natural decline of individuals. Individual’s lifetime equals one iteration.

Figure 8 J-shaped population growth model.

Propagation of individuals occurs in the absence of intraspecific competition and any restrictions on the resources. A species colonizes an infinite ecosystem under ideal conditions. (A) The number of offsprings per individual equals three. (B) The number of offsprings per individual equals six. (C) Geometric population growth in the first case (A). (D) Geometric population growth in the second case (B).

To assess the effect of intraspecific competition and regeneration of microhabitats on population growth, we have compared our model of the S-shaped (Fig. 7A and Movie S5) with the J-shaped population growth (Figs. 8B and 8D). Comparative dynamics of these models is shown in Table 1. Comparison of these two examples shows that intraspecific competition is a powerful factor which limits population growth. We also compared our double S-shaped population growth model (Fig. 7C and Movie S7) with the J-shaped population growth model (Figs. 8A and 8C). Comparative dynamics of these models is shown in Table 2. Thus, we have compared our models of the S-shaped and the double S-shaped population growth with the J-shaped population growth.

Table 1 Comparative population dynamics in the models with the S-shaped and the J-shaped population growth.

Time (Number of iteration and generation)	0	1	2	3	4	5	
Number of individuals in the S-shaped population growth model (Fig. 7A and Movie S5). Intraspecific competition exists. Fecundity equals 6 individuals.	1	6	13	24	37	54	
Number of individuals in the J-shaped population growth model (Figs. 8B and 8D). Intraspecific competition is absent. Geometric population growth. Fecundity equals 6.	1	6	36	216	1,296	7,776	

Table 2 Comparative population dynamics in the models with the double S-shaped and the J-shaped population growth.

Time (Number of iteration and generation)	0	1	2	3	4	5	
Number of individuals in the S-shaped population growth model (Fig. 7C and Movie S7). Intraspecific competition exists. Fecundity equals 3.	1	3	6	10	15	21	
Number of individuals in the J-shaped population growth model (Figs. 8A and 8C). Intraspecific competition is absent. Geometric population growth. Fecundity equals 3.	1	3	9	27	81	243	

The basic ecological model, which has been presented in this paper, can easily be expanded by the introduction of additional states, different neighborhoods, nested and adjoint lattices (Kalmykov & Kalmykov, 2012). Such extension has allowed us to create pure mechanistic models of interspecific competition between two, three and four species that are complete competitors, and then to verify and reformulate the competitive exlusion principle (Kalmykov & Kalmykov, 2013) in order to solve the biodiversity mystery (Sommer, 1999).

Conclusions

We have presented and investigated a mechanistic model of dynamics of single species plant population. This model is based on pure logical deterministic individual-based cellular automata. It has a physical and ecological ontology. Here the physical ontology is the ontology of the active medium and ecological ontology represents an ecosytem with one vegetatively propagated plant species. We investigated deterministic individual-based mechanisms underlying the S-shaped and double S-shaped population growth of vegetatively propagated plants. Imitating modeling of vegetatively propagated rhizomatous lawn grasses was not our main goal. The main goal was demonstration of possibilities of the white-box modeling on example of population growth. The white-box model was made on the basis of physical axioms of excitation propagation in excitable medium. These basic physical axioms of the model have a universal character that, in principle, allows transferring the obtained results to other subject areas. An additional important result is itself demonstration of the white-box modeling of complex systems using logical cellular automata. We consider the details of the “white-box modeling” methodology as the main results of our work. We would like to make this perspective approach more widely used in the practice of mathematical modeling of complex systems. And we have tried to supplement the discussion about “the value of white boxes” by considering specific ways of implementation this model approach. Our study directly introduces the white-box approach into ecological modeling. The white-box approach opens up new perspectives in modeling by implementing a multy-level mechanistic modeling of complex systems. Our deterministic logical cellular automata model works as a system of artificial intelligence. Cellular automata are known as the method of artificial intelligence. But there is a problem how to use this method for investigation of complex systems. We show how logical deterministic cellular automata may be used for mathematical white-box modeling of complex systems on example of ecosystem with one species. Parallelism of the logical operations of cellular automata in total volume of the modeled macrosystem allows to speak that the model hyper-logically provides automatic deductive inference. The term ‘deductive’ is used here because all logical operations are based on axioms. We consider that the main difficulty of this white-box modeling is to create an adequate axiomatic system based on an intrinsic physical ontology of the complex system under study. The main feature of the approach is the use of cellular automata as a way of linking semantics (ontology) and logic of the subject area. Our white-box model of an ecosystem with one species combines reductionist and holistic approaches to modeling of complex systems. We consider the white-box modeling by logical deterministic cellular automata as a perspective way for investigation of any complex systems.

Supplemental Information

Movie S1 Monte Carlo simulation. Torus-lattice. Hexagonal neighborhood

Every Monte Carlo simulation consisted of 100 repeated experiments with different initial positioning of an individual on the lattice. Here are shown five repeated experiments. The lattice is uniform, homogeneous and limited. It consists of 23 × 23 sites available for occupation by individuals. The lattice is closed on the torus and the neighborhood is hexagonal.

Click here for additional data file.

Movie S2 Monte Carlo simulation. The lattice has a boundary. Hexagonal neighborhood

Every Monte Carlo simulation consisted of 100 repeated experiments with different initial positioning of an individual on the lattice. Here are shown five repeated experiments. The lattice is uniform, homogeneous and limited. It consists of 23 × 23 sites available for occupation by individuals. The lattice has a boundary and the neighborhood is hexagonal.

Click here for additional data file.

Movie S3 Monte Carlo simulation. Torus-lattice. Tripod neighborhood

Every Monte Carlo simulation consisted of 100 repeated experiments with different initial positioning of an individual on the lattice. Here are shown five repeated experiments. The lattice is uniform, homogeneous and limited. It consists of 23 × 23 sites available for occupation by individuals. The lattice is closed on the torus and the neighborhood is tripod.

Click here for additional data file.

Movie S4 Monte Carlo simulation. The lattice has a boundary. Hexagonal neighborhood

Every Monte Carlo simulation consisted of 100 repeated experiments with different initial positioning of an individual on the lattice. Here are shown five repeated experiments. The lattice is uniform, homogeneous and limited. It consists of 23 × 23 sites available for occupation by individuals. The lattice has a boundary and the neighborhood is tripod.

Click here for additional data file.

Movie S5 S-shaped population growth with short phase of decelerating growth

The lattice size which is available for occupation consists of 50 × 50 sites. Cellular automata neighborhood is hexagonal and the lattice is closed on the torus.

Click here for additional data file.

Movie S6 S-shaped population growth with sharp transition to long phase of decelerating growth

The lattice size which is available for occupation consists of 50x50 sites. Cellular automata neighborhood is hexagonal and the lattice has a boundary.

Click here for additional data file.

Movie S7 Double S-shaped population growth

The lattice size which is available for occupation consists of 50x50 sites. Cellular automata neighborhood is tripod and the lattice is closed on the torus.

Click here for additional data file.

Movie S8 S-shaped population growth with very long deceleration phase

The lattice size which is available for occupation consists of 50x50 sites. Cellular automata neighborhood is tripod and the lattice has a boundary.

Click here for additional data file.

We thank Kylla M. Benes for helpful suggestions and edits of the earlier version of the manuscript. We would like to thank the Academic Editor Markus Dahlem and the anonymous reviewers for fruitful comments.

Additional Information and Declarations

Competing Interests

Author Contributions

The authors declare there are no competing interests.

Lev V. Kalmykov conceived and designed the experiments, performed the experiments, analyzed the data, contributed materials/analysis tools, wrote the paper, prepared figures and/or tables, reviewed drafts of the paper, made programs and movies.

Vyacheslav L. Kalmykov conceived and designed the experiments, analyzed the data, contributed materials/analysis tools, wrote the paper, prepared figures and/or tables, reviewed drafts of the paper.

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
