# Peer review of "A white-box model of S-shaped and double S-shaped single-species population growth"

_PeerJ, doi:10.7717/peerj.948_

## Round 0.1 · original submission · Major Revisions

Unfortunately, we were waiting quite some time for a second reviewer, who had accepted the invitation. Since we did not receive their review, I think we should and can proceed. We also had many other reviewers who all declined to review the paper.

During the several times I contacted this second reviewer, he actually provided me with an assessment saying that the manuscript follows a very generic approach and by his reading, he questioned what concrete results have emerged.

To a certain degree, this is in line with the reviewer #1. And at this point let me suggest that you follow carefully the various points brought up by reviewer #1.

Reviewer 1 ·

Basic reporting

The paper describes a “white-box” approach to modeling population balances. Specifically, the paper presents a study of rhizomatous lawn grasses under various model assumptions. The modeling work is interesting but the manuscript in its present form has several deficiencies and I cannot recommend publication.

* The title “population growth” is very generic. It is used widely, including many areas outside the scope of the paper, for example, in chemistry and physics (as in population balances, polymerization, colloidal systems etc). I suggest adding to the title an indication of the specific problem that is discussed.

* On the question of white/black-box models. I found the discussion a bit repetitive, perhaps because I don’t need much convincing as to the value of white boxes. However, the authors should also comment on a difficulty with such models: If the mechanistic details are unknown, as is often the case, white models risk becoming computer games without relevance for the real problem.

* The authors state “A pure mechanistic model of a complex system is a discrete logical model consisting of cause effects relations and part-whole relations”, but I do not understand what is mean by “part-whole” relationships.

* The “micro/meso/macro” levels mentioned in lines 159-162 are unclear.

* The statement (line 162-3) “Interactions between subsystems of a complex system lead to emergence of its new properties” is generic, and unsubstantiated.

* The authors find that initial positioning under periodic conditions does not change the dynamics. That’s exactly what is expected. Why is this important to report?

* The authors do not motivate the study of periodic versus not periodic boundaries. The standard practice is to use periodic boundaries in order to eliminate edge effect. Why are non-periodic boundaries included? Are the authors seeking to quantify finite-size effects?

* I have trouble with statements such as “We consider this holystic multi-level white-box modeling approach as a method of artificial intelligence which works as hyper-logical automatic deductive inference that provides direct mechanistic insights into complex systems.”

What does that mean?

Or “It [the model] hyper-logically provides automatic deductive inference. “

I cannot make sense of these statements in the context of mathematical modeling.

Experimental design

No comments

Validity of the findings

No comments

---

## Round 0.2 · accepted · Accept

You made a significant effort to handle the points raised by the reviewer. The abstract got a bit long that way. I would suggest you shorten it a bit. For example, the first sentence seems too general and can be cut out ("A mechanistic modeling of complex systems is one of the main scientific problems today"). Currently, you have 357 words. I suggest you aim at 250 words -- keeping the specific and most important information.